# COPD Guidelines in the Asia-Pacific Regions: Similarities and Differences

**DOI:** 10.3390/diagnostics11071153

**Published:** 2021-06-24

**Authors:** Shih-Lung Cheng, Ching-Hsiung Lin

**Affiliations:** 1Department of Internal Medicine, Far Eastern Memorial Hospital, Taipei 22060, Taiwan; 2Department of Chemical Engineering and Materials Science, Yuan Ze University, Zhongli District, Taoyuan 320315, Taiwan; 3Division of Chest Medicine, Department of Internal Medicine, Changhua Christian Hospital, Changhua 50006, Taiwan; teddy@cch.org.tw; 4Institute of Genomics and Bioinformatics, National Chung Hsing University, Taichung 40227, Taiwan; 5Department of Recreation and Holistic Wellness, MingDao University, Changhua 50006, Taiwan

**Keywords:** COPD guideline, Asia

## Abstract

Chronic obstructive pulmonary disease (COPD) is a preventable and treatable disease that is associated with significant morbidity and mortality, giving rise to an enormous social and economic burden. The Global Strategy for the Diagnosis, Management and Prevention of Chronic Obstructive Pulmonary Disease (GOLD) report is one of the most frequently used documents for managing COPD patients worldwide. A survey was conducted across country-level members of Asia-Pacific Society of Respiratory (APSR) for collecting an updated version of local COPD guidelines, which were implemented in each country. This is the first report to summarize the similarities and differences among the COPD guidelines across the Asia-Pacific region. The degree of airflow limitation, assessment of COPD severity, management, and pharmacologic therapy of stable COPD will be reviewed in this report.

## 1. Introduction

Although chronic obstructive pulmonary disease (COPD) is a preventable and treatable disease, it is associated with significant morbidity and mortality, giving rise to an enormous social and economic burden. The results from the Epidemiology and Impact of COPD (EPIC) Asia population-based survey suggest a high prevalence of COPD in the participating Asia-Pacific territories [1] and indicate a substantial socioeconomic burden of the disease in this region. Individuals with the disease reported substantial limitations in their daily activities and loss in work productivity. To address this situation and influence the behavior of healthcare providers and health policy makers and payers, numerous organizations have developed clinical practice guidelines (CPG) to assist in the diagnosis and treatment of COPD. In such an environment, CPG development often relies upon expert opinion. Conflicting interpretations of the literature regarding COPD management may result in disparities across guidelines. Local factors, such as the availability of certain health care services or the cost impact of an intervention, may also influence how local experts view and apply the published literature during guideline development.

The Global Strategy for the Diagnosis, Management and Prevention of Chronic Obstructive Pulmonary Disease (GOLD) report is one of the most frequently used documents for managing COPD patients worldwide [2,3]. It was developed by using an evidence-based methodology and expert opinion consensus and is considered the most up-to-date, comprehensive reference for COPD diagnosis and management. However, a major gap is that its focus is only in the application of the recommended GOLD strategies for pharmacological treatment of COPD based on the A, B, C, and D groups. Here, our focus of this survey is to determine the degree of consensus in the Asian Pacific region’s practice guidelines for COPD regarding the diagnosis and management of COPD.

### Estimated Prevalence

The prevalence of COPD in the Asia-Pacific countries is estimated at 14.5% in Australia [4], 4.4% to 16.7% in China [5,6,7], and 5.6% in Indonesia [1]; the prevalence of Air Flow Limitation (FEV1/FVC < 70%) was reported at 10.9% and COPD (after excluding asthmatics) was 8.6% in Japan [8], 13.4% in Korea [9], 4.7% in Malaysia [1], 5.4% to 6.1% in Taiwan [1,10], 3.7% to 6.8% in Thailand [11], 3.5% to 20.8% in Philippines [1,12], and 6.7% in Vietnam [1], respectively (Table 1).

## 2. Method

A survey was conducted across country-level members of the Asia-Pacific Society of Respiratory (APSR) for collecting an updated version of COPD guidelines which were implemented in each country. The APSR sent a questionnaire to members, who were asked to provide the current local guideline and comparative review of the collected guidelines. Ten guidelines were reviewed, including those of Australia/New Zealand, China, Indonesia, Japan, Korea, Malaysia, Taiwan, Thailand, Philippines, and Vietnam, in either English or national language. The key disease management graphs, flowcharts, and algorithms were translated into English language for review. Detailed information was completely collected, including the definition, the approach to diagnosis, severity classification of staging, pharmacotherapy for stable COPD, and other recommendations. In the Asia-Pacific available COPD guidelines, Australia, Japan, Korea, Taiwan, and China have revised and updated guidelines during the period of 2013 to 2020 (Table 1). Guidelines in the other countries were not revised in the recent three years. We compared the similarities and differences between these guidelines.

The different methods used to estimate disease prevalence including expert opinion, patient-reported diagnosis, and symptom-based or spirometry-based methods may affect the results. In the People’s Republic of China, COPD is one of the most common chronic diseases in the population older than 40 years of age, with a prevalence of 8.2% in 2007 and increased to 13.6% in 2015 using spirometry-based survey. [5,7] Comparatively higher prevalence with 13.7% to 13.4% was noted in Korea using spirometry-based survey [9,13]. Another study in the Asia-Pacific region, EPIC Asia population-based survey [1] based on face-to-face or fixed-line telephone interviews, revealed that the prevalence of COPD is between 6.2% and 19.1%. Regarding the estimated prevalence rate of COPD in each country, there is no appropriate method to do this in current status.

## 3. Results

COPD diagnosis, classification, and treatment recommendation from Taiwan and China were similar to the GOLD guidelines. The degree of airflow limitation, assessment of COPD severity, management, and pharmacologic therapy of stable COPD were based on the GOLD principles. Australia, Japan, and Korea guidelines display some differences regarding classification and management strategy of stable COPD compared with the GOLD (Table 2). Besides, Taiwan guidelines have been written based on GRADE (Grading of Recommendations, Assessment, Development and Evaluations)’s recommendation, which is the most widely adopted tool for grading the quality of evidence and for making recommendations.

### 3.1. Combined COPD Assessment

The Korean COPD guideline categorizes severity into three groups, Group ga (GOLD Group A), Group na (GOLD Group B), and Group da (GOLD Group C and D) [13] (Figure 1). The spirometric cutoff point of FEV1 is 60% predicted to distinct Group ga, na from Group da. They further divide Group da into two groups with FEV1 < 60% predicted, but >=50% predicted, or FEV1 < 50% predicted. [14]. Assessment of symptoms and exacerbation is similar as described in GOLD. In Australia, COPD-X concise guide [15] for primary care categorizes the severity of COPD into mild (FEV1: 60–80% of predicted), moderate (FEV1: 40–59% of predicted), and severe (FEV1: <40% of predicted) accompanied with typical symptoms of varying degree of dyspnea, cough, and limitation of daily activity (Figure 2) [16]. The rationale was that regular treatment with inhaled corticosteroid (ICS) can improve symptoms, lung function, quality of life, and reduce the frequency of exacerbation for patients with FEV1 < 50% predicted and a history of frequent exacerbations, observed in several clinical studies [16,17,18].

### 3.2. Pharmacologic Management of Stable Disease

In the GOLD guideline, the initial pharmacological management of COPD is according to patient group which has different recommended treatments. In the guidelines of Australia, Japan, and Korea (Figure 2 [15], Figure 3 [19], and Figure 4 [14]), a stepwise approach of optimized pharmacotherapy for stable COPD is used which recommends a gradual increase of bronchodilators, inhaled corticosteroids, or other drugs based on a comprehensive evaluation of symptoms, airflow obstruction, and exacerbation. In Japan’s 2018 guideline, ICS positioning for COPD treatment had been revised from the previous criteria of FEV1 < 50% of predicted, frequent exacerbation, and concomitant asthma to only the concomitant asthma (ACO) criterion.

### 3.3. Non-Pharmacologic Management

Most guidelines had emphasized the importance of pulmonary rehabilitation, long-term oxygen therapy, and self-management plan including smoking cessation and vaccination. Particularly, Japan’s guideline (fifth edition) discussed the nutrition management including nutritional impairment, evaluation, therapy, and diet education [19]. COPD patients whose BMI is less than 90% are suspected to have a nutrition disorder and nutrition therapy may be indicated. Nutritionists, physician, and nurses should form a team to provide nutritional guidance.

### 3.4. Coexisting Asthma and COPD

Coexisting asthma and COPD are only defined and described in Australia and Japan guidelines. This Australia guideline recommends that an FEV1 increase over 12% and 200 mL constitutes a positive bronchodilator response. An FEV1 increase >400 mL strongly suggests underlying asthma or coexisting asthma and COPD diagnosis. Besides, the diagnosis of asthma–COPD overlap (ACO) has both characteristics of COPD and asthma (Figure 5).

### 3.5. End-of-Life ISSUES

GOLD 2013, for the first time, proposed that palliative care may be applied in advanced severe COPD patients. Among these guidelines in the Asia-Pacific region, Taiwan, Japan, China [20], and Australia [15] may already have their policies about end-of-life care. Improving quality of life, optimizing function, helping with decision- making about end-of-life care, and providing emotional and spiritual support to patients and family are the main goals. In Taiwan, the National Health Insurance Administration Ministry of Health and Welfare had programmed hospice-care plans in 2011 and provided in-hospital critical care facilities for patients with advanced diseases and poor response to regular treatments instead of home or hospice ward care.

## 4. Discussion

There are several studies evaluating and validating the new GOLD assessment system; however, uneven distribution of COPD patients and limited data on the clinical outcomes are noticed under these combined assessments. [21,22,23,24] The degree of the COPD Assessment Test (CAT) score of ≥10 might not be equivalent to that of the mMRC score of ≥2 for categorizing patients’ symptoms. [25,26,27,28] Neither the 2007 GOLD nor the 2011 classification scheme has sufficient discriminatory power to be used clinically for risk classification to predict total mortality at the individual level. [29] Accordingly, some countries have developed COPD guidelines to build up appropriate strategies for diagnosis, assessment, pharmacotherapy, and prediction of acute exacerbation and mortality based on evidence and real-world clinical practice.

The Korean and Australia guidelines stratified the lung function severity and exacerbation risk with FEV1 < 60% or ≥ 60% of predicted value. From the validation study in Korea, it was found that there were many patients (15.3% to 16%) who experienced exacerbation with FEV1 between 50% and 60% of predicted value. [14] The cutoff point of an FEV1 50% predicted does not address the heterogeneity in the GOLD Stage II (50%–80% predicted). Patients with limited airflow around FEV1 50% to 60% predicted had a more rapid decline in lung function than patients with FEV1 < 50% in the TORCH study [30,31]. A recent study showed that parameters related to volume, diffusing capacity, and reactance showed break-points around 65% of FEV1 which may have an impact on patients’ management plan.

The strategy for stable COPD management was based on lung function severity before GOLD 2011. A refinement of the ABCD assessment tool had been separated from spirometric grade from “ABCD” groups in GOLD 2020. A stepwise approach policy is currently presented in the Japan and Australia guidelines. The management strategy is similar in the Korea and GOLD guidelines including for symptoms severity and exacerbation frequency. Moreover, a phenotype-guided treatment policy has been shown in the Spanish and Czech guidelines. [32,33] Which strategies are optimal in clinical practice guidelines for COPD management? There were several strategies including lung function-guided, stepwise approach-guided, GOLD A–D-guided, and phenotype-guided strategies. The optimal treatment of COPD patients requires an individualized, multidisciplinary approach to the lung function severity, patient’s symptoms, clinical phenotypes, biomarkers, comorbidity evaluation, and needs.

The treatment of patients with COPD in a more personalized way must address diverse aspects not only related with the disease, but also with its comorbidities, and current schemes do not offer such personalized medical treatment. Comorbidity evaluation and management were all mentioned in each Asia country CPG. In the JRS guideline l19], the comorbidities included systemic inflammation, osteoporosis, musculoskeletal defect, cardiovascular disorders, gastro-intestinal dysfunction, depression, metabolic disorders, and obstructive sleep apnea. Additionally, the variability of the clinical presentation interacts with comorbidities to form a complex clinical scenario for clinicians. Different comorbidities have different evaluation and management policies. Consequently, the CPG or consensus should be reached over a practical approach for combining comorbidities and disease presentation markers in the therapeutic algorithm, in order to improve the quality of clinical care.

In a previous study, the increased total health expenditure was shown as share GDP ≥ 7% in Korea, Japan, and Australia in 2007. [34] In Japan, major reforms are needed to reduce waste and enhance cost-effectiveness. Moreover, a national system to accredit training programs, including for general practice, has been introduced. [35] The challenges of the healthcare system in Korea include over-consumption and excessively high frequency of specialist consultation, which are major problems for the medical system. The government and the primary care group seek to strengthen primary care, but this is opposed by the medical society governed by the specialist group. [35] In Australia, some provider payment methods were performed such as case payment, diagnostic-related groups, etc. [34]. We think that guideline differences are driven by the disparities in diagnosis modalities or by the treatment variations in different healthcare systems and the socioeconomic burden in each country.

Additionally, diagnosis tools and management of COPD were among the lower guideline-recommended levels in most of the regions investigated among primary care physicians or general practitioners (GPs). [36] The survey demonstrated that the GPs’ understanding of COPD was variable and large numbers of GPs have very limited knowledge of COPD and its management in Asia countries. The percentage for COPD management by guideline is as follows: Australia 64%, Japan 74%, Korea 54%, and Taiwan 70%. In China, only 50% of patients with COPD have ever had spirometry tests in tertiary hospitals, and only 18% had in primary or secondary hospitals. [37] Therefore, from the education system, clinical practice, and medical impact, there appears to be an optimal strategy developed to simplify the guidelines for daily practice in each country.

Research evidence has raised concerns that hospital death may be preceded by potentially burdensome and inappropriate hospital admission and aggressive treatments shortly before death, which could be a threat to better end-of-life care and death. [38,39,40,41] On the other hand, enabling people to have end-of-life care at home compared with end-of-life care in hospital may incur a potential cost saving. [42,43] The concepts of palliative and hospice care should be established gradually in regards to diseases with an advanced stage.

### APSR Recommendations for COPD Diagnosis and Treatment

COPD is characterized by persistent respiratory symptoms and airflow limitation. Spirometry is required to make the diagnosis.The severity of COPD should be comprehensively assessed on the basis of the degree of obstruction severity (FEV1, GOLD stage), impairment of exercise tolerance/physical activity, intensity of dyspnea, and frequency/ severity of exacerbation.The goal of pharmacological treatment should be to treat the symptoms (e.g., breathlessness) or to prevent deterioration (either by decreasing exacerbations or by reducing the decline in lung function and quality of life) or both. A stepwise approach is recommended, irrespective of disease severity, until adequate control has been achieved.Management of non-pharmacological strategies for stable COPD should center around supporting smoking patients to quit. Encouraging physical activity and maintenance of a normal weight range are also important. Pulmonary rehabilitation is recommended in all symptomatic patients.Stepwise management of optimized pharmacotherapy for stable COPD which recommends gradual increase of bronchodilators, inhaled corticosteroids, or other drugs based on clinical symptoms, airflow obstruction severity, and exacerbation history.ICS should be used in cases with concomitant asthmatic conditions and/or 2 or more exacerbations in the previous 12 months. LABA/ICS combinations are also allowed.In the end-of-life care, improving quality of life and providing emotional and spiritual support to COPD patients and their family are the main goals.

## 5. Conclusions

This is the first report to summarize the similarities and differences among the COPD guidelines across the Asia-Pacific region. The guideline developed in each country would be based on clinical evidence, experts’ consensus, healthcare insurance, reality of clinical practice, and the best interests of patients. We hope, through collaboration of research, that the guidelines will evolve positively and that differences or gaps will diminish with time.

## Figures and Tables

**Figure 1 diagnostics-11-01153-f001:**
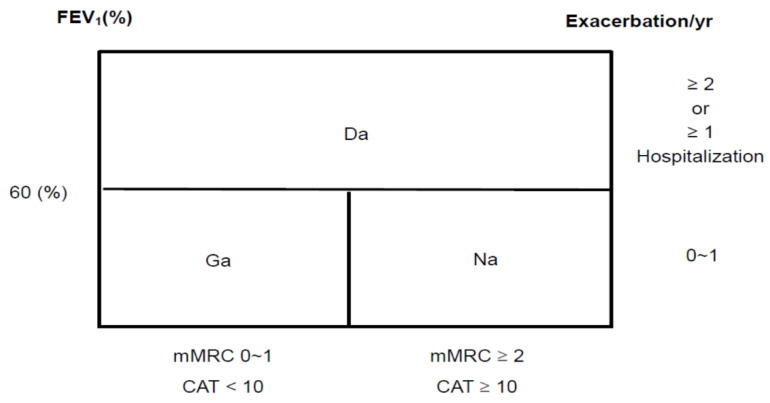
Korean COPD classification system and GOLD classification system.

**Figure 2 diagnostics-11-01153-f002:**
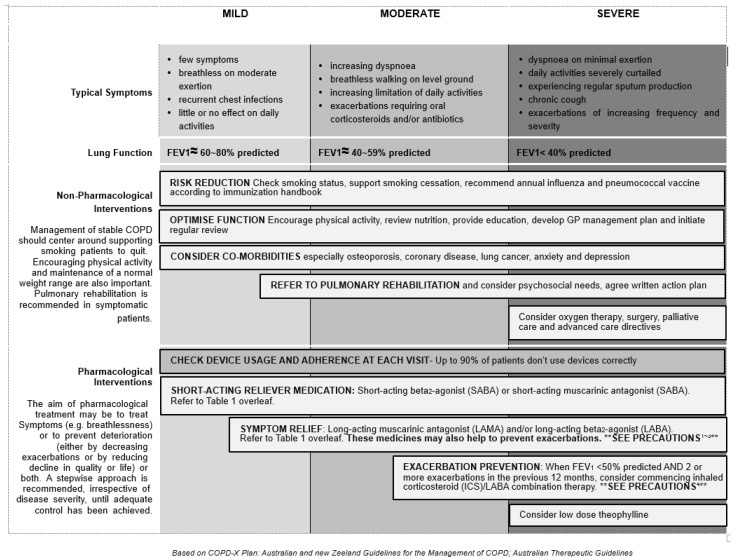
Stepwise management of stable COPD guidelines in Australia and New Zealand.

**Figure 3 diagnostics-11-01153-f003:**
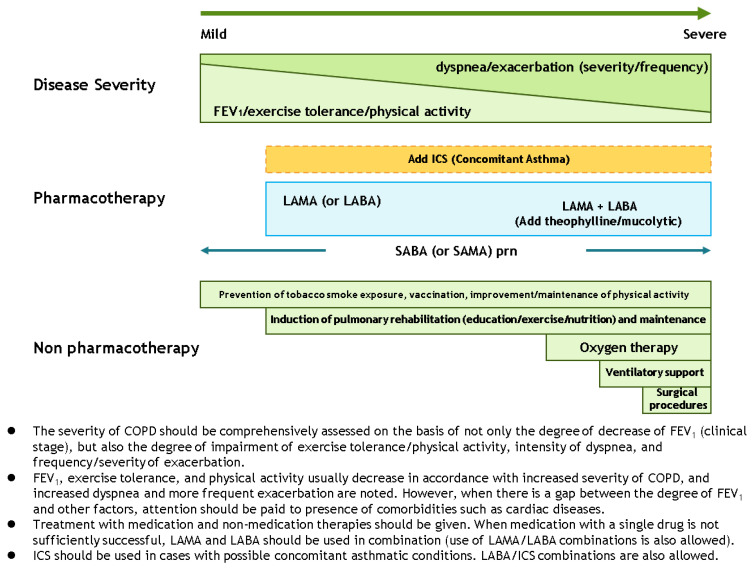
Stepwise approach recommended by the fifth edition of Japanese Respiratory Society COPD guidelines.

**Figure 4 diagnostics-11-01153-f004:**
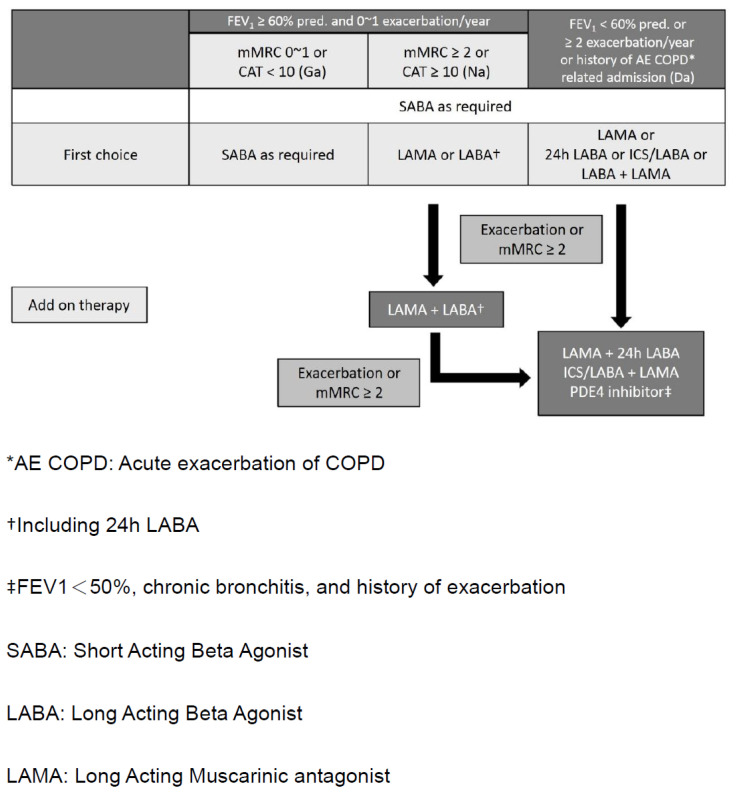
Algorithm of pharmacologic treatment in patients with stable COPD in Korea.

**Figure 5 diagnostics-11-01153-f005:**
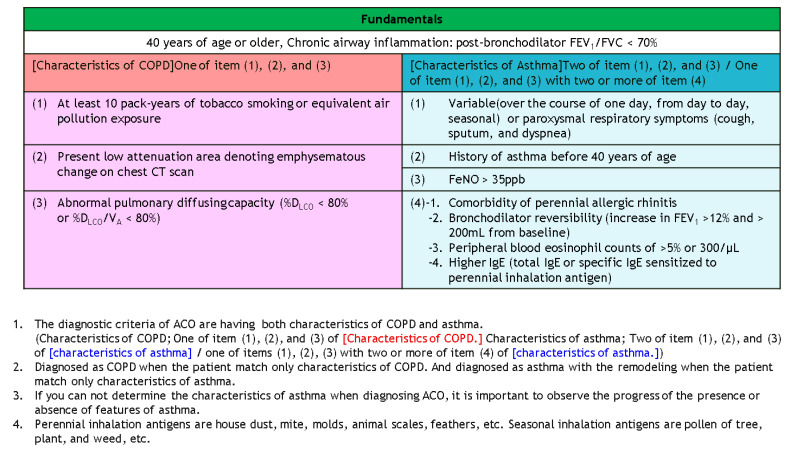
Diagnostic criteria of asthma–COPD overlap in Japanese ACO guideline 2018.

**Table 1 diagnostics-11-01153-t001:** Publication year of current and last version of Asia Pacific (APAC) guidelines, and COPD prevalence in the reviewed APAC countries.

	Australia/New Zealand *	China	Indonesia	Japan	Korea	Malaysia	Taiwan	Thailand	Philippines	Vietnam
Current version	2020	2017	2011	2018	2018	2009	2020	2010	2009	2009
Last version	2013	2007	NA	2009	2014	1998	2011	NA	2003	2009
Planned next version	NA	NA	NA	NA	NA	NA	2023	2016	NA	2018
COPD prevalence	14.5%	4.4–16.7%	5.6%	8.6–10.9%	13.4%	4.7%	5.4–6.1%	3.7–6.8%	3.5–20.8%	6.7%

* Stepwise management table of COPD was published in 2017; Concise Guide for Primary Care (COPD-X plan) was published in 2017.

**Table 2 diagnostics-11-01153-t002:** Comparison of GOLD 2015 and APAC guidelines with current version updated after 2011.

	Disease Classification and Management Recommendation Same as GOLD	Major Difference in COPD Diagnosis Classification	Major Difference in COPD Treatment Recommendation
Australia	No	(1) Typical symptoms and lung function assessed in parallel for COPD severity classification (2) FEV1 40%, 60% and 80% predicted as the cut points of COPD severity (3) No specified cut points of mMRC and CAT for symptom evaluation	(1) Stepwise management of stable COPD; therapeutic choices appropriately fully aligned with disease severity.
China	Yes		
Japan	No	(1) Typical symptoms and lung function assessed in parallel for COPD severity classification (2) No specified cut points of mMRC and CAT for symptom evaluation	(1) Stepwise management of stable COPD; therapeutic choices not fully aligned with disease severity.
Korea	No	(1) FEV1 60% predicted as the cut point of high- and low-risk class. (2) Combined GOLD C and GOLD D into one group (Korean group ‘da’)	(1) Specified criteria, the occurrence of exacerbation or mMRC ≥2 despite of current treatment, for add up treatment from first therapeutic choice. (2) Mixed treatment recommendation of GOLD C and D for group ‘da.’
Taiwan	Yes		

## Data Availability

Not applicable.

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
