# Peer review of "COPD Guidelines in the Asia-Pacific Regions: Similarities and Differences"

_diagnostics, 2021, doi:10.3390/diagnostics11071153_

Round 1

Reviewer 1 Report

In Japan, 1st version was 1999, and 2nd was 2004, then 3rd was 2009, 4th was 2013, and current was 2018. 

My major concern on current version is in Discussion section, around Line 175, they using “?” and mentioned several comments but this style looks odd. Moreover, they dis not describe more about comorbidity in this manuscript. They only said as "Which strategy is optimal in clinical practice still debated? Optimal treatment of COPD patients requires an individualized, multidisciplinary approach to the lung function severity, patient’s symptoms, clinical phenotypes, biomarkers, comorbidities evaluation and needs.” I really want to see more description especially Japanese guideline will be good to refer in these comments. In addition I think various guideline, were not referred appropriately (ref 19 was 4th version. not 5th). Moreover, Japanese guideline was originally in Japanese. I think it could be better to say who translated the schema from guideline. In management of COPD patients, comorbifdity should be considered appropriately. I do suggest describe more about comorbitidies in the present review. In minor point, there are ton of typo or style errors.

Author Response

Dear Reviewer 1:

The manuscript has been revised according to Reviewer 1 and 2’ suggestion. We have added the discussion and description as your comments. Additionally, we would reply the questions point by point.

We deeply appreciated the expert comment and considerate recommendation from you. We would like to thank you for your thorough review and excellent suggestion. Thanks again for reviewers’ kindness and give us the opportunity to revise our manuscript. Hope to have good news for acceptance.

Best regards

Shih-Lung Cheng, MD. PhD

Department of Internal Medicine, Far Eastern Memorial Hospital, Taipei, Taiwan

Phone: 886-2-89667000 ext 2160      

Fax: 886-2-77380708        

Dear Reviewer 1:

Thanks for the review’s excellent recommendations and suggestions.

I will reply these suggestions point by point:

In Japan, 1st version was 1999, and 2nd was 2004, then 3rd was 2009, 4th was 2013, and current was 2018.

Authors reply: Thanks for your comments, We have revised JRS in 2018 to 5th version.

My major concern on current version is in Discussion section, around Line 175, they using “?” and mentioned several comments but this style looks odd. Moreover, they dis not describe more about comorbidity in this manuscript. They only said as "Which strategy is optimal in clinical practice still debated? Optimal treatment of COPD patients requires an individualized, multidisciplinary approach to the lung function severity, patient’s symptoms, clinical phenotypes, biomarkers, comorbidities evaluation and needs.” I really want to see more description especially Japanese guideline will be good to refer in these comments. In addition I think various guideline, were not referred appropriately (ref 19 was 4th version. not 5th). Moreover, Japanese guideline was originally in Japanese. I think it could be better to say who translated the schema from guideline. In management of COPD patients, comorbifdity should be considered appropriately. I do suggest describe more about comorbitidies in the present review. In minor point, there are ton of typo or style errors.

Authors reply: Thanks for your comments. We have revised and added the description about comorbidities in the Discussion section. We would revise the reference 19 to 5th JRS COPD guideline in 2018.

Besides, we have revised our English grammar and style errors.

Reviewer 2 Report

The authors provided the review artcle to this journal.

Several concerns have raised.

Comment1.

The authors aimed to write a literature review on the importance of accurate and timely COPD diagnosis. If this is the case then the manuscript’s structure should be reorganized in this direction, following the PRISMA guidelines for systematic reviews.

Comment2.

A Methodology section needs to be included, where to state that this is a literature review and present the databases searched, key words used, criteria for paper inclusion and exclusion, as well as the whole process algorithm including the PRISMA flow chart for systematic reviews.

Comment3.

What is the new information that this study adds synopsis and interpretation of the findings expanding on their significance define the problem.

Author Response

Dear Reviewer 2:

The manuscript has been revised according to Reviewer 1 and 2’ suggestion. We have added the discussion and description as your comments. Additionally, we would reply the questions point by point.

We deeply appreciated the expert comment and considerate recommendation from you. We would like to thank you for your thorough review and excellent suggestion. Thanks again for reviewers’ kindness and give us the opportunity to revise our manuscript. Hope to have good news for acceptance.

Best regards

Shih-Lung Cheng, MD. PhD

Department of Internal Medicine, Far Eastern Memorial Hospital, Taipei, Taiwan

Phone: 886-2-89667000 ext 2160      

Fax: 886-2-77380708        

Dear Reviewer 2:

Thanks for the review’s excellent recommendations and suggestions.

I will reply these suggestions point by point:

Several concerns have raised.

Comment1.

The authors aimed to write a literature review on the importance of accurate and timely COPD diagnosis. If this is the case then the manuscript’s structure should be reorganized in this direction, following the PRISMA guidelines for systematic reviews.

Author reply: Thanks for your comments. The study aims to enroll clinical practical guidelines (CPG) in Asia countries and compared the similarities and differences. We thought that the study was neither literature review nor systemic review. The study was not suitable for PRISMA guidelines for systemic reviews.

Comment2.

A Methodology section needs to be included, where to state that this is a literature review and present the databases searched, key words used, criteria for paper inclusion and exclusion, as well as the whole process algorithm including the PRISMA flow chart for systematic reviews.

Author reply: Thanks for your comments. This is the first report to summarize the similarities and differences among the COPD guidelines across the Asia-Pacific region. The degree of airflow limitation, assessment of COPD severity, management and pharmacologic therapy of stable COPD will be reviewed in this report. The report was neither a literature review nor systemic review. Therefore, we would not add PRISMA flow chart for systemic review.

Comment3.

What is the new information that this study adds synopsis and interpretation of the findings expanding on their significance define the problem.

Author reply: Thanks for your comments. This report wanted to summarize the similarities and differences among the COPD guidelines (CPG) across the Asia-Pacific region. The guideline developed in each country would be based on clinical evidence, experts’ consensus, health care insurance, reality of clinical practice and the best interests of patients. We hope, through collaboration of research, the guideline will evolve positively and differences or gap will diminish with time. We believed that the report would let readers to understand the similarities and differences, which advantages and disadvantages should be adjusted by readers. There were no update information in the study, because it was very difficult to revise or update clinical practice guidelines in each countries in a short time.

Round 2

Reviewer 2 Report

I feel they would not respond to my comments.

The authors also mentioned that there were no update information in the study.

Author Response

Dear Reviewer 2:

Thanks for the review’s excellent recommendations and suggestions.

I will reply these suggestions point by point:

The authors also mentioned that there were no update information in the study.

Author reply: We have updated the available information in the study and revised the references.
